# Van der Waals pressure and its effect on trapped interlayer molecules

K.S. Vasu[1], E. Prestat[2], J. Abraham[1], J. Dix[3], R.J. Kashtiban[4], J. Beheshtian[5], J. Sloan[4], P. Carbone[3], M. Neek-Amal[5], S.J. Haigh[2], A.K. Geim[1] & R.R. Nair[1]

Van der Waals assembly of two-dimensional crystals continue attract intense interest due to the prospect of designing novel materials with on-demand properties. One of the unique features of this technology is the possibility of trapping molecules between two-dimensional crystals. The trapped molecules are predicted to experience pressures as high as 1 GPa. Here we report measurements of this interfacial pressure by capturing pressure-sensitive molecules and studying their structural and conformational changes. Pressures of 1.2 ± 0.3 GPa are found using Raman spectrometry for molecular layers of 1-nm in thickness. We further show that this pressure can induce chemical reactions, and several trapped salts are found to react with water at room temperature, leading to two-dimensional crystals of the corresponding oxides. This pressure and its effect should be taken into account in studies of van der Waals heterostructures and can also be exploited to modify materials confined at the atomic interfaces.

[1] School of Physics and Astronomy, University of Manchester, Manchester M13 9PL, UK. [2] School of Materials, University of Manchester, Manchester M13 9PL, UK. [3] School of Chemical Engineering and Analytical Science, University of Manchester, Manchester M13 9PL, UK. [4] Department of Physics, University of Warwick, Coventry CV4 7AL, UK. [5] Department of Physics, Shahid Rajaee Teacher Training University, 16875-163, Lavizan, Tehran, Iran. Correspondence and requests for materials should be addressed to R.R.N. (email: rahul@manchester.ac.uk).

Van der Waals (vdW) interactions play a critical role in numerous phenomena and applications such as catalysis, adhesion, lubrication, nanofluidics and fabrication of novel vdW materials[1–10]. In the latter case, two-dimensional (2D) crystals assembled layer by layer are kept together only by vdW forces[1,11]. Various heterostructures composed of graphene, BN, $MoS_2$ and so on have been intensively investigated and already shown potential for a wide range of electronic and optical applications[3,4]. During the fabrication of such heterostructures, molecules become trapped between 2D crystals, which results in enclosures of nanometre height[5]. Such nano-enclosures were also fabricated intentionally, mostly using graphene, to serve as hydrothermal reactors[12,13], visualize chemical processes[14,15] and for electron and atomic force microscopy (AFM) of biological samples and adlayers[6,16,17]. Hydrothermal reactions reported inside the graphene nanobubbles were limited to the pressure created at high temperatures, where trapped solvents and molecules reach their critical or super critical phase to initiate the chemical reactions similar to the conventional hydrothermal anvil cell[13]. When nano-enclosures are created at room temperature, a pressure $P$ builds up, and it can be extremely high due to an exceptionally high stiffness of graphene and the fact that vdW forces become extremely strong on a sub-nanometre scale. The value of $P$ is determined by a balance between the resulting hydrostatic pressure, the deformation energy of 2D crystals and a gain in the adhesion energy[6,18]. The deformation energy favours enclosures with a flat top. In the first approximation, the pressure can be estimated as $\approx E_w/h$, where $E_w$ is the adhesion energy and $h$ is the height of the nano-enclosure[6,18]. For a trapped monolayer, this estimate yields $P$ of the order of 1 GPa, in agreement with further rigorous analysis[7,18,19]. However, such large vdW pressure has never been measured experimentally, nor has its effect on physical, structural and chemical properties of entrapped molecules been studied.

Here we demonstrate the existence of the huge vdW pressure inside the nano-enclosures made from 2D crystals by studying changes in Raman spectra of pressure-sensitive molecules such as triphenyl amine (TPA) and boric acid (BA). We also report the effect of vdW pressure on the chemistry of materials inside nano-enclosures by investigating the chemical stability of trapped compounds [$MgCl_2$, CsI, $CuSO_4$ and $Ca(OH)_2$] using both Raman spectroscopy and transmission electron microscopy (TEM).

## Results

**Fabrication of nano-enclosures.** Various 2D crystals have been used in our experiments to prepare nano-enclosures as described in detail in the Methods and Supplementary Fig. 1. For brevity, we focus below on enclosures made from an archetypal 2D material, graphene. Graphene-encapsulated (GE) molecule/salt structures were prepared by drop casting a small amount (2 μl) of a dilute ($\leq 0.1$ M) molecular or salt solution onto a graphene layer. A second (top) graphene crystal was then carefully placed on top using the standard transfer techniques[1,3] to trap the solution. We normally used few-layer graphene as a bottom layer to minimize its crumpling during the drop casting. The liquid between graphene crystals was allowed to dry overnight at room temperature and then further dried in a vacuum desiccator. As a result, the two graphene layers strongly attached to each other with a tiny amount of the solution being captured in between.

**vdW pressure measurements.** Figure 1a shows an optical image of one of our GE-TPA samples. One can see many small bubbles (indicated with arrows in Fig. 1a) within seemingly featureless areas. However, AFM reveals that these areas are actually composed of flat nano-enclosures with a height of $\approx 1$ nm (region 1 in the inset of Fig. 1a), which are surrounded by numerous submicron bubbles of several nanometres in height (region 2). Such 1-nm-thick flat regions are found to be continuous, and their lateral size can be as large as several micrometres. A similar AFM landscape was found for GE-BA samples (Supplementary Note 1 and Supplementary Fig. 2a).

To determine the pressure inside this type of graphene nano-enclosure, we perform Raman spectroscopy. Figure 1b compares spectra obtained from a bubble of several microns in size and from a flat 1-nm-thick region. The Raman spectrum for bubble agrees with the known spectrum of bulk TPA[20] where the major Raman bands are centred at $\approx 997$ cm$^{-1}$ due to C–C stretching accompanied by weak C–N stretching, 1,025 cm$^{-1}$ due to C–N stretching with associated C–C stretching and 1,172 cm$^{-1}$ due to combined C–N and C–C stretching. In stark contrast, the Raman spectrum of flat GE-TPA regions show all the bands being blueshifted with peaks appearing at $\approx 1,011$, 1,040 and 1,195 cm$^{-1}$, respectively. The widths of these bands are larger than those of bulk TPA. The observed changes in the Raman bands of captured TPA molecules are attributed to structural

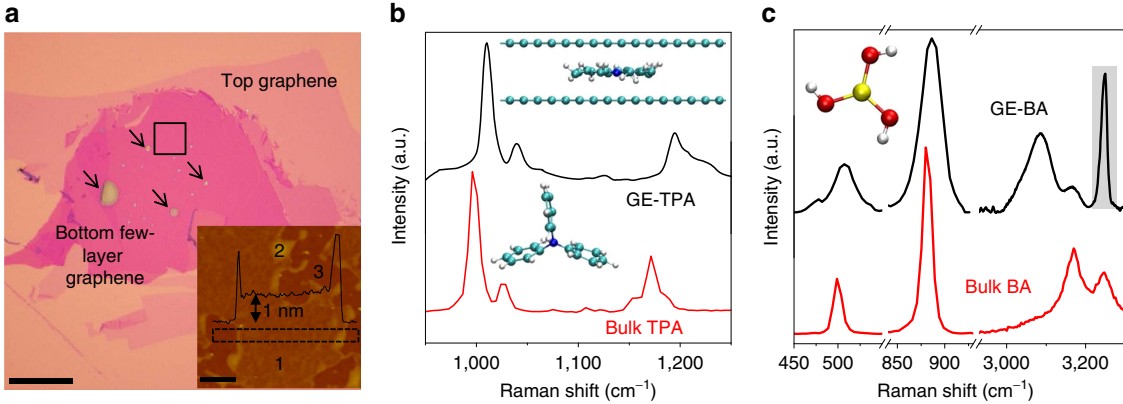

**Figure 1 | Probing vdW pressure by Raman spectroscopy. (a)** Optical micrograph of a graphene-encapsulated TPA on few-layer graphene placed on an oxidized Si wafer. Scale bar, 30 μm. Arrows indicate some of the small bubbles in the sample. Inset: AFM image of the region indicated by the square in **a**. A 1-nm-thick flat region is marked as 1, one of many small bubbles marked as 2 and the region in which top and bottom graphene are attached as 3. Scale bar, 1 μm. Black curve: height profile along the dotted rectangle. **(b)** Typical Raman spectra (514 nm excitation) from macroscopic bubbles and regions such as 1. The bottom inset shows the standard molecular structure of TPA. Top inset: modified structure of TPA between graphene sheets as found in MD simulations. **(c)** Raman spectra (514 nm excitation) of bulk BA and GE-BA. Inset: molecular structure of BA. The shaded area indicates the 2D′ band of graphene.

changes because of the confinement of TPA between graphene layers (Fig. 1b). Indeed, the propeller-shaped structure of TPA makes it highly sensitive to the uniaxial confinement, which can be translated into uniaxial pressure on the molecule. The observed Raman changes are in good agreement with those found for TPA at high pressures using conventional measurements in diamond anvil cells[21]. Comparison of our results with the latter measurements allows an estimate of $P$ inside our flat graphene nano-enclosures as 1–1.5 GPa. To support this conclusion, we have also performed molecular dynamic simulations for TPA confined in a graphene capillary. They show that TPA molecules undergo a conformational change (from the propeller to planar structure; inset in Fig. 1b), if the separation between graphene sheets decreases below 1 nm (Supplementary Note 2 and Supplementary Figs 3–5).

Similar experiments were performed with boric acid. Unlike TPA, BA molecules have a nearly planar structure already at ambient pressure. They are attracted to each other by hydrogen bonding and organize layered structures[22]. Figure 1c shows Raman spectra obtained from bulk BA and from a 1-nm-thick enclosure with BA. The main bands for bulk BA are at 500 cm$^{-1}$ that corresponds to O–B–O angle deformation, 880 cm$^{-1}$ for B–O stretching, 3,167 cm$^{-1}$ for symmetric O–H stretching and 3,245 cm$^{-1}$ for antisymmetric O–H stretching[23]. For 1-nm-thick encapsulated BA, the two B–O modes exhibit broadening and a blueshift (509 and 885.5 cm$^{-1}$, respectively), whereas the O–H stretching modes show a redshift and a notable change in the shape of the peaks (Fig. 1c). Similar to the case of TPA, the observed changes in the Raman spectrum of flat GE-BA are consistent with previously reported hydrostatically pressurized BA[23]. Distinct from the other Raman bands of BA, the redshift for O–H stretching vibrations under pressure can be attributed to shortening of the hydrogen bond, which leads to lengthening and weakening of the parent O–H bond under compression[24]. The structure of BA is highly sensitive to pressure so that the O–B–O deformation and symmetric B–O stretching modes shift by 7.0 and 4.5 cm$^{-1}$ per GPa, respectively[23]. This allows us to estimate the pressure inside our 1-nm-thick graphene enclosures as 1.2 ± 0.3 GPa, in good agreement with the vdW pressure measured for TPA. Besides probing the vdW pressure, Raman spectroscopy also allowed us to estimate the strain developed in top-layer graphene during the BA/TPA

encapsulation by analysing the shift in peak position of graphene G and 2D bands (Supplementary Note 3 and Supplementary Fig. 6).

**Effects of vdW pressure on trapped salts.** The extremely high vdW pressure acting on substances captured between 2D crystals cannot be ignored and may modify physical and chemical properties of both trapped compounds and confining crystals experiencing the same pressure. As an example, we show that compounds such as $MgCl_2$, $CuSO_4$ and $Ca(OH)_2$ become reactive with water at room temperature under the pressure inside nano-enclosures. Figure 2a shows an AFM image of $MgCl_2$ solution trapped between graphene layers (Supplementary Note 1 and Supplementary Fig. 2b,c). Similar to the case of trapped TPA and BA, GE-$MgCl_2$ samples contain numerous flat regions with $h \approx 1$ nm. Figure 2b plots typical Raman spectra acquired from such regions for 0.1 M solutions of $MgCl_2$ in water and methanol. As a reference, we also provide Raman spectra from an anhydrous $MgCl_2$ powder and a region where a large droplet of 0.1 M aqueous solution of $MgCl_2$ was allowed to dry up on top of graphene outside the nano-enclosures.

Anhydrous $MgCl_2$ exhibits two Raman bands, one at $\approx 240$ cm$^{-1}$ due to out-of-plane vibrations and another at $\approx 157$ cm$^{-1}$ assigned to in-plane phonons[25]. Also, the Raman spectrum obtained from the dried $MgCl_2$ solution shows typical signatures of hydrated $MgCl_2$ (ref. 26; Supplementary Note 4 and Supplementary Fig. 7) with a broad redshifted peak at $\approx 200$ cm$^{-1}$. In contrast, the Raman spectrum of 0.1 M aqueous $MgCl_2$ inside the flat GE region does not show any of the expected features but instead a sharp peak appears at $125 \pm 3$ cm$^{-1}$. This new Raman band was universally observed for many flat encapsulated regions, reaching several microns in size. In addition to the 125 cm$^{-1}$ band, we also observe changes in the Raman spectrum of graphene covering $MgCl_2$ regions, which include the emergence of the D peak (Supplementary Note 5 and Supplementary Fig. 8). In comparison, nano-enclosures with $MgCl_2$ dissolved in methanol show only broadened Raman signature characteristic of anhydrous $MgCl_2$ and no sign of the 125 cm$^{-1}$ peak (Fig. 2b).

Although the described spectroscopic changes indicate marked changes in the structure of the trapped aqueous $MgCl_2$, Raman spectroscopy is unable to reveal what kind of transformation

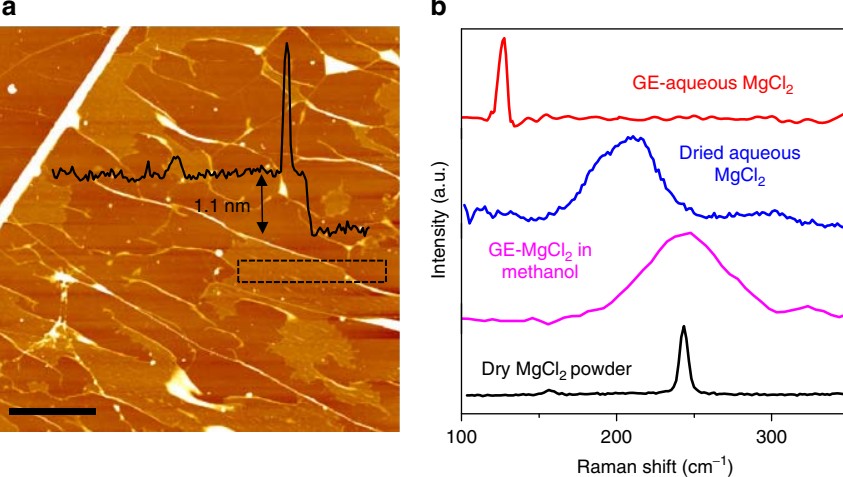

**Figure 2 | Encapsulation-induced chemical reactions.** (**a**) AFM image obtained from graphene-encapsulated aqueous $MgCl_2$. Scale bar, 2 μm. Nanometre-thick flat regions are clearly seen being separated by wrinkles and occasional bubbles of triangular and arbitrary shapes. Black curve: height profile along the dotted rectangle. (**b**) Raman spectra (488 nm excitation) measured from flat regions encapsulating $MgCl_2$ dissolved in water and methanol. For comparison, spectra for anhydrous and hydrated $MgCl_2$ are shown.

takes place. To elucidate this, we have performed TEM analysis of constituents inside the nano-enclosures (see Methods). Our low-magnification TEM images confirm that the samples contain flat enclosed areas with only a few small bubbles in between (inset of Fig. 3a), in agreement with our AFM imaging. Elemental analysis using energy dispersive X-ray (EDX) spectroscopy shows that the bubbles contain magnesium, oxygen and chlorine, as expected for aqueous $MgCl_2$, whereas the flat enclosures contain magnesium and oxygen but surprisingly with no signature of chlorine (Fig. 3b). High-resolution TEM also reveals that the flat regions are thin crystallites of one to few layers in thickness, which have a square lattice with lattice constant of $2.1 \pm 0.05$ Å (Fig. 3a). Electron diffraction and electron energy loss spectroscopy (EELS) analyses confirm that these nanocrystals are MgO, with the EELS oxygen K-edge from the enclosed material showing a good match to the reference spectrum for bulk MgO (Fig. 3c). Furthermore, fast Fourier transform of high-resolution image of the thinnest MgO crystallites show the presence of a (110) lattice reflection, and our simulated electron diffraction patterns confirm that such (110) reflections are a clear signature of monolayer MgO (Supplementary Note 6 and Supplementary Figs 9 and 10). All this unequivocally shows that the observed nanocrystals inside graphene nano-enclosures are MgO. This conclusion is also consistent with the observed $125\,cm^{-1}$ Raman band that was previously reported for MgO crystals[27,28].

## Discussion

We have performed similar experiments with aqueous solutions of other compounds including $CuSO_4$, CsI and $Ca(OH)_2$ (Supplementary Note 7 and Supplementary Figs 11 and 12). Similar to the case of $MgCl_2$, our Raman and TEM analyses show that graphene-encapsulated $CuSO_4$ and $Ca(OH)_2$ are converted into CuO and CaO, respectively, whereas graphene-encapsulated CsI remains stable. This agrees with our density functional theory calculations performed for $MgCl_2$, $Mg(OH)_2$ and CsI trapped with water inside a hydrophobic cage (Supplementary Note 8 and Supplementary Fig. 13). They suggest the formation of a Mg–O

bond in the case of $MgCl_2$ and $Mg(OH)_2$, and no changes for CsI. Although the calculations do not elucidate the underlying mechanism for the room-temperature reactivity of the salts with $H_2O$, it is reasonable to assume that the reactions are induced by the high vdW pressure. Indeed, the probability of ionization of water molecules strongly increases with increasing $P$, and changes in pressure from 0.1 MPa to 1 GPa are known to cause the ionization constant of water to change by two orders of magnitude[29], thus favouring the hydrolysis. We propose that the observed room-temperature reaction of nano-enclosed salts with water involves the pressure-assisted hydrolysis of salts into their hydroxides followed by decomposition to the corresponding oxides (Supplementary Note 9). Similar vdW pressure-assisted reactions were also observed in other 2D crystal nano-enclosures such as BN (Supplementary Note 10 and Supplementary Fig. 14). Conversion of bulk salts to corresponding hydroxides or oxides was only observed previously at high temperatures and pressures[30], and we assume that the confinement and pressure has a significant influence on the feasibility of observed reaction at room temperature. The effect of confinement apparently brings the close proximity of metal ions to ionized water molecules to begin hydrolysis reaction and it is important to note that without the vdW confinement the pressure alone could not make the observed reaction feasible at room temperatue[31,32]. The reaction byproducts such as HCl and water are probably accumulated in the small bubbles and wrinkles observed in the flat encapsulated regions (Supplementary Fig. 2b) or evaporated through microscopic defects created in graphene during the reaction as the presence of graphene's D peak indicates (Supplementary Note 5 and Supplementary Fig. 8). The proposed mechanism is also consistent with the absence of changes for CsI. In this case, the salt of strong acid and strong base completely ionizes in water rather than undergoing hydrolysis[33]. In contrast, $MgCl_2$ and $CuSO_4$ are examples of salts of a weak base and strong acid that normally undergo hydrolysis. The observed non-reactivity for $MgCl_2$ in methanol is naturally explained by the absence of water. Still, further experimental and theoretical studies are needed for a better understanding of the chemical transformations inside graphene nano-enclosures.

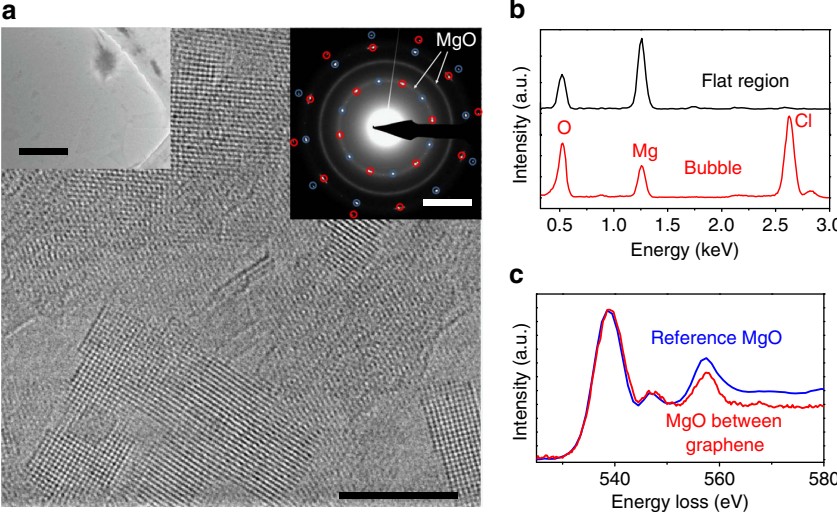

**Figure 3 | TEM on graphene-encapsulated aqueous $MgCl_2$.** (**a**) High-resolution TEM image of MgO nanocrystal formed between two graphene layers. Scale bar, 5 nm. Inset on left of panel: low-magnification TEM image showing flat encapsulated regions and a larger bubble. Scale bar, 500 nm. Inset on right of panel: diffraction pattern taken from a 1-μm-diameter flat area shows two hexagonal patterns (red and blue circles), which come from the top and bottom graphene layers, and additional diffraction rings corresponding to the {200} and {220} planes of polycrystalline MgO. Scale bar, 5 nm$^{-1}$. (**b**) EDX spectra from a flat encapsulated region and from a large bubble. (**c**) Oxygen-K edge spectrum from nanocrystals such as in **a** compared with the spectrum of bulk MgO from the EELS database (https://eelsdb.eu/spectra/magnesium-oxide-2).

In conclusion, our work shows that huge pressures are exerted on materials trapped at interfaces during assembly of vdW heterostructure and that these pressures can lead to unexpected physical and chemical changes. This should be considered in analysis of the properties of vdW heterostructures, especially because the neighbouring 2D crystals experience the same pressure. Our results can be exploited to induce novel physical phenomena in nano-enclosures and create new chemical compounds, which all points at many opportunities to explore the science of nanoconfined materials.

## Methods

**Fabrication of graphene-encapsulated samples.** Graphene-encapsulated samples were prepared by using the wet transfer technique[34] to enclose salt or molecule solutions between two micromechanically exfoliated graphene flakes or between a graphene flake and a graphite flake as summarized in Supplementary Fig. 1. Single- and few-layer graphene flakes were prepared on Si/SiO2 substrates with an oxide thickness of 300 nm using the micromechanical exfoliation[35]. Optical microscopy was used to locate the graphene flakes of suitable size ($50 \times 50\,\mu m$ or above) on Si/SiO2 substrate (panel 1 in Supplementary Fig. 1) and the thickness of these flakes was verified using Raman spectroscopy[36]. A thin layer ($\approx 200\,nm$) of A3-950 poly(methyl methacrylate) (PMMA) resist was then spin coated on to a substrate containing the desired flake and subsequently heated at 100 °C for 10 min to evaporate the solvent from the resist (panel 2 in Supplementary Fig. 1). An adhesive tape window was placed above the PMMA layer, ensuring that the desired graphene flake was at the centre of the open area (panel 3 in Supplementary Fig. 1). The tape and PMMA act as a support for the graphene flake allowing it to be lifted away from the Si/SiO2 substrate. PMMA layer at the periphery of the tape window was then removed to separate the tape window area from the remainder of the PMMA layer. The entire sample was then placed in 3% KOH solution to etch away the SiO2 layer and thereby separate the graphene flake from the substrate (panel 4 in Supplementary Fig. 1). After this etching process, the tape window with the PMMA layer containing the graphene flake floated on the surface of the KOH solution due to the hydrophobic nature of PMMA, whereas the Si substrate stays at the bottom (panel 5 in Supplementary Fig. 1). Subsequently, the tape window hosting the graphene–PMMA layer was rinsed with deionized water to remove any residual KOH solution. Using the micromanipulation set-up[37], this graphene–PMMA layer was carefully placed on top of 2 μl of the molecule or salt solution casted onto the bottom few-layer graphene or graphite flake. We have used 0.001–0.1 M solutions of the molecule or salt, and all reported results were similar even when the different concentrations of molecule or salt solutions were used. All sample solutions were aqueous, except TPA, which is insoluble in water, so methanol was used as the solvent. A single-layer graphene (SLG) has also been tested as a bottom layer but was not ideal as it often ruptured during drop casting of the molecule or salt solution. After placing the single layer of graphene on top of the solution, most of the solution was spontaneously squeezed out by leaving only a very small amount in between the top and bottom graphene layers. The prepared samples were dried overnight at room temperature to let the solvent to evaporate gradually and allow the top graphene layer to collapse onto the bottom graphene flake. These samples were then placed in vacuum ($\sim 1\,mBar$) for several hours to remove any residual solvent and water molecules. Finally, the tape window was detached, and the top PMMA layer was removed using an acetone wash.

Graphene grown using chemical vapour deposition on Cu foil, received from BGT Materials Limited (Manchester, UK), has been used to prepare samples suitable for TEM and scanning TEM high-resolution imaging and analysis. To detach the chemical vapour deposition grown graphene from the Cu foil ($5 \times 5\,mm$), the graphene was coated with PMMA and the metal was etched away using a 0.1-M ammonium persulfate aqueous solution. The resulting graphene–PMMA film was then rinsed several times in water and transferred onto a gold mesh Quantifoil TEM support grid. Removal of the PMMA layer was achieved by washing in acetone and isopropyl alcohol baths. The graphene-covered TEM grids produced were then dried in a critical point dryer and subsequently annealed in activated carbon at 250 °C to remove the hydrocarbon contamination[38]. To prepare the GE salt samples for TEM, salt solution was drop casted at the centre of a graphene-covered TEM grid followed by placing another graphene-covered TEM grid above the droplet. The prepared TEM samples were further dried as described above.

**AFM measurements.** AFM imaging of the encapsulated samples was performed using a Bruker Dimension FastScan AFM operating in peak force tapping mode.

**Raman measurements.** A micro Raman set-up consisting of an optical microscope, and a spectrograph with 2,400 and 1,800 lines per mm grating (Reinshaw/Witec spectrometer) was used to obtain Raman spectra in a back-scattering geometry using a $\times 100$ objective lens (numerical aperture = 0.70) under ambient conditions. An Ar-ion multiline tuning laser operating at wavelengths of 488 and 514 nm and a He–Ne laser operating at a wavelength of 633 nm were used as the excitation sources for the Raman measurements in this study. We chose a laser power well below 2 mW to avoid any laser-induced heating effects during the measurements. For the Raman experiments, thick graphite flakes were used as the bottom layer in the encapsulated samples to avoid the background Raman spectrum of the Si/SiO2 substrate.

**TEM imaging.** Aberration-corrected TEM, EDX spectroscopy and electron diffraction experiments were carried out using a JEOL ARM 200F TEM microscope operated at an operating voltage of 80 kV. The dose rate for high-resolution transmission electron microscope imaging was maintained at $10^4$ electrons per Å$^2$ per s. EDX spectroscopy was performed using the ARM's window-less Oxford Instruments X-MaxN 100TLE detector and TEM images were recorded using a Gatan Orius charge-coupled device camera.

An FEI Titan 80–200 ChemiSTEM equipped with probe-side aberration correction and an X-FEG electron source was used for the aberration-corrected scanning transmission electron microscope (STEM) imaging, EELS and electron diffraction analysis. STEM experiments were performed using an acceleration voltage of 200 kV, a beam current of 150 pA and a convergence angle of 21 mrad. EDX spectrum imaging was performed using the Titan's four detector Super-X spectrometer system, providing a solid angle of $\sim 0.7\,sr$. EELS data were acquired using a GIF Quantum ER Spectrometer, with an energy dispersion of 0.25 eV and a collection angle of 62 mrad, providing an effective energy resolution of 1.8 eV. In standard STEM imaging mode, the high-angle annular dark field collection angles were 62–142 mrad, while during spectrum imaging the high-angle annular dark field collection angles were 60–190 mrad. Electron diffraction patterns were acquired with a beam current of 2 nA and with an illuminated area on the specimen of $\approx 1\,\mu m^2$.

**Data availability.** The data that support the findings of this study are available from the corresponding author on request.

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

## Acknowledgements

This work was supported by the Royal Society and the Engineering and Physical Sciences Research Council, UK (EP/K016946/1). S.J.H. and E.P. acknowledge support from the Defense Threat Reduction Agency (HDTRA-12-01-0013). This work was also supported by SuperSTEM: the U.K.'s National Facility for aberration corrected STEM funded by the Engineering and Physical Sciences Research Council.

## Author contributions

R.R.N. designed the project and directed it with help from K.S.V.; K.S.V. prepared the samples, performed the measurements and carried out data analysis; E.P, J.A., R.J.K., J.S. and S.J.H. carried out electron microscopy measurements and data analysis; J.D., J.B., P.C. and M.N.-A. carried out DFT calculations and molecular dynamic simulations; A.K.G. participated in discussions and project design; R.R.N., K.S.V. and A.K.G. wrote the manuscript. All authors contributed to discussions.

## Additional information

**Competing financial interests:** The authors declare no competing financial interests.

