## [Peer Review File · Nature Communications]

REVIEWERS' COMMENTS:

Reviewer #1 (Remarks to the Author):

The authors have addressed many of the previous concerns and highlighted the difference of their work to previous work, and have revised and updated their references.

The main selling point is the fact that even at room temperature, the nano 2D bubbles create a highly pressurized environment of 1 GPa, as can be seen from the entrapment of pressure sensing molecules (from the work of Candy et. al. in *Angewandte*, now cited in this version), whereas the work reported previously by Candy required heating to raise the pressure to GPa regimes.

The interesting outcome is that even at room temperature, room temperature chemical conversion of metal ions can occur, even when this temperature is highly prohibitive under normal conditions, which suggests that the ultrahigh pressure is playing a role.

The authors can address the question of whether the earlier observation of ice formation in trapped G layers is an example of "pressure induced solidification", is this a different manifestation of such room temperature induced solidification reactions. Perhaps under such pressurised 2D confinement, there is a higher tendency for ions to be reduced and to nucleate on graphene. Bulk reaction enthalpies consideration did not consider the role of a confined 2D substrate, top and bottom, for lowering the activation barrier for nucleation

This version of the paper has been improved and can be considered for publication

Reviewer #2 (Remarks to the Author):

The authors have submitted a revised manuscript which addresses several of the points raised in the previous version. Except for the reaction energy and intercalation related effects, the rest of the concerns have been addressed. After addressing and/or acknowledging the possibility of reaction energy or intercalation-like formation, the manuscript may be published in Nature Communications.

1. The role of interactions has been included.
2. MD simulations have been conducted; although results are not included.
3. The concern on the possibility of reaction with salts was not been addressed well. The intercalation of metal salts in graphite and BN is well known. However, it was not discussed in the manuscript. Only citing a paper on intercalation (15) indicating it only as an example of chemical visualization is insufficient. As a starting point, the observed lattice spacing and distances can be compared with Mg, Ca, Cs, etc intercalated graphite and BN.

We thank all referees for careful reading of our manuscript. We find their comments helpful, and all the suggestions are incorporated in the revised manuscript as specified below.

Reply to Comments of Referee 1

Reviewer #1 (Remarks to the Author):

The authors have addressed many of the previous concerns and highlighted the difference of their work to previous work, and have revised and updated their references. The main selling point is the fact that even at room temperature, the nano 2D bubbles create a highly pressurized environment of 1 GPa, as can be seen from the entrapment of pressure sensing molecules (from the work of Candy et. al. in *Angewandte*, now cited in this version), whereas the work reported previously by Candy required heating to raise the pressure to GPa regimes. The interesting outcome is that even at room temperature, room temperature chemical conversion of metal ions can occur, even when this temperature is highly prohibitive under normal conditions, which suggests that the ultrahigh pressure is playing a role.

We thank the referee for this assessment.

The authors can address the question of whether the earlier observation of ice formation in trapped G layers is an example of "pressure induced solidification", is this a different manifestation of such room temperature induced solidification reactions. Perhaps under such pressurised 2D confinement, there is a higher tendency for ions to be reduced and to nucleate on graphene.

We thank the referee for his/her interest in our previous work.

Room-temperature chemical reactions reported in the present paper are different from the square ice found between graphene layers. Molecular dynamic simulations show that even though van der Waals pressure may be important to form highly crystalline square lattice for trapped water there is no indication of a pressure-induced solidification. Even in the absence of high pressure, trapped water is expected to form ice crystals due to strong confinement (ref. 6 in this manuscript).

Bulk reaction enthalpies consideration did not consider the role of a confined 2D substrate, top and bottom, for lowering the activation barrier for nucleation

We do agree with the referee and acknowledged the need for dedicated enthalpy calculations for assessing this effect further in the revised supplementary information (page 22, supplementary note 9).

This version of the paper has been improved and can be considered for publication

We thank the referee for this kind assessment.

Reply to Comments of Referee 2

Reviewer #2 (Remarks to the Author):

The authors have submitted a revised manuscript which addresses several of the points raised in the previous version. Except for the reaction energy and intercalation related effects, the rest of the concerns have been addressed. After addressing and/or acknowledging the

possibility of reaction energy or intercalation-like formation, the manuscript may be published in Nature Communications.

1. The role of interactions has been included.

2. MD simulations have been conducted; although results are not included.

We thank the referee for this kind assessment. The revised manuscript now addresses both referee concerns. We have included a new paragraph acknowledging the possibility of reaction energy calculation for understanding the exact reaction mechanism in the revised supplementary note 9 (page 22).

Regarding the intercalation related effects, we have now included the relevant explanations (see below) in the revised supplementary note 5 (Page 18).

3. The concern on the possibility of reaction with salts was not been addressed well. The intercalation of metal salts in graphite and BN is well known. However, it was not discussed in the manuscript. Only citing a paper on intercalation (15) indicating it only as an example of chemical visualization is insufficient. As a starting point, the observed lattice spacing and distances can be compared with Mg, Ca, Cs, etc intercalated graphite and BN.

We do agree with the referee that certain salts (e.g. FeCl_3) and metals (Cs, Ca etc.) form intercalated compounds with graphite or BN, however, intercalation typically occurs at high temperature in the presence of molten salts or metals or via electrochemical techniques. Due to the absence of these conditions in our experiments we don't expect such intercalation under our experimental conditions. In addition, intercalation of salts or metals increases the charge carrier concentration in graphene which can easily be identified by Raman spectroscopy of graphene. The absence of any notable blue shifts in the G-peak position indicates the absence of intercalation in our samples. We have added this information in the revised supplementary note 5. In addition, our electron spectroscopy and diffraction analysis clearly showed that the enclosed materials inside the nano-enclosures are two-dimensional crystals of metal oxide rather than intercalated salts or metals.